# Frequency-Hopping Binary Offset Carrier Modulation with Independent Frequency-Hopping Patterns in Lower and Upper Sidebands

**Yikang Yang** [1],*, **Jiangang Ma** [2], **Lingyu Deng** [1] **and Hengnian Li** [3]

1 School of Electronic and Information Engineering, Xi'an Jiao Tong University (XJTU), Xi'an 710049, China; dengly0625@stu.xjtu.edu.cn
2 Key Laboratory of Science and Technology on Space Microwave, China Academy of Space Technology, Xi'an 710100, China; jiangangma@126.com
3 State Key Laboratory of Astronautic Dynamics, Xi'an Satellite Control Center, Xi'an 710049, China; henry_xscc@mail.xjtu.edu.cn
* Correspondence: yangyk74@mail.xjtu.edu.cn

**Abstract:** To enhance the anti-interference capabilities and increase flexibility in frequency allocation between the lower and upper sidebands of the navigation signal, we introduce frequency-hopping binary offset carrier modulation with independent frequency-hopping patterns in lower and upper sidebands (IFH-BOC). This novel modulation is classified as a constant-envelope multiplexing (CEM) method, with independent frequency-hopping patterns for the lower and upper sidebands, in contrast with frequency-hopping binary offset carrier (FH-BOC) and binary offset carrier (BOC) modulations, which share the same patterns. IFH-BOC represents a generalized modulation that incorporates FH-BOC and BOC, thus retaining their advantages while introducing new characteristics, such as independent frequency-hopping pattern design and flexible spectral splitting. The results indicate that IFH-BOC maintains the same time–frequency characteristics and measurement accuracy as FH-BOC when using identical modulation parameters, yet it demonstrates superior anti-interference performance due to its varied frequency-hopping patterns. Furthermore, IFH-BOC provides enhanced flexibility in spectral splitting compared with BOC modulation, potentially allowing for increased availability of L-band frequencies for satellite navigation. With these benefits, IFH-BOC is poised to be a promising modulation for the signal design of next-generation global navigation satellite systems.

**Keywords:** IFH-BOC; independent frequency-hopping patterns; anti-interference; navigation





## 1. Introduction

As the demand for large-scale, multi-type location-based services increases, global navigation satellite systems (GNSS) must accommodate various services by transmitting distinct navigation signals [1,2]. Binary offset carrier (BOC) modulation was developed to reduce interference between new and legacy signals while enhancing positioning accuracy. This technique has been widely implemented in GNSS [3–5]. The BOC signal is generated by multiplying a binary phase shift keying (BPSK) signal with rectangular spreading symbols and a binary offset square wave subcarrier, which divides the power spectral density (PSD) of the BPSK signal into lower and upper sidebands at negative and positive frequencies of the BOC subcarrier, respectively. While the autocorrelation function (ACF) of the BPSK signal displays a single triangular main lobe, the ACF of the BOC signal includes multiple triangular side peaks along with the main lobe.

The advancement of GNSS has fostered the ongoing development of BOC modulation, leading to various BOC-like modulations. Several BOC-like techniques have been introduced to optimize satellite payload utilization and enhance spectral efficiency, which

also function as constant-envelope multiplexing (CEM) methods. One notable example is alternated BOC (Alt-BOC), a dual-frequency CEM approach implemented for the Galileo E5 signal [6]. Multiple enhancements to Alt-BOC were proposed during the development of the BeiDou Navigation System (BDS) [7–14]. These CEM techniques, derived from BOC modulation, successfully combine various BOC signals into a single constant-envelope composite signal, thereby improving both spectral efficiency and design flexibility in GNSS signals. The unique characteristics of the ACF and PSD of BOC signals enable enhanced spectral isolation from legacy signals, leading to greater accuracy, improved resistance to multipath interference, and increased flexibility in signal implementation compared with BPSK-R signals. However, they also introduce challenges, such as ACF peak ambiguity, which can result in significant measurement bias [15–17]. Generalized BOC (GBOC) modulation was proposed to address this ambiguity, which features a subcarrier frequency that changes linearly and continuously within each spreading code chip [18]. GBOC effectively minimizes ACF side peaks, reducing the likelihood of incorrect code phase acquisition and the chances of false code locks during tracking. Additionally, Ma et al. introduced frequency-hopping BOC modulation (FH-BOC), achieved by combining the BPSK signal with rectangular spreading symbols and a frequency-hopping binary offset square wave subcarrier [19]. FH-BOC alleviates the ACF side peak ambiguity associated with BOC signals while significantly enhancing anti-interference capabilities, thus improving positioning accuracy in challenging environments. Kong et al. proposed stepped-frequency BOC (SFBOC) modulation, which produces a BOC signal with a subcarrier frequency that changes incrementally during each spreading code period [20]. The SFBOC signal exhibits a sharp ACF and demonstrates resilience to code Doppler effects. Moreover, Deng et al. introduce a wideband multi-carrier navigation modulation, which uses an orthogonal BOC signal as a subcarrier and has good compatibility with wideband communication signals [21].

The advancement of BOC modulation has enabled the constant-envelope multiplexing of various signals, reduced the ambiguity associated with side peaks of the ACF, and enhanced anti-interference capabilities. However, the inherent spectrum-splitting characteristic of BOC signals remains static, allowing only a symmetrical shift of the signal energy to the lower and upper subcarrier frequencies around the radio frequency. This limitation restricts the flexibility of GNSS signal design. Additionally, the increasingly congested L-band, with its restricted bandwidth, presents a challenge, as fewer available frequencies for navigation may lead to interference with legacy signals [22]. It is essential to enhance the flexibility of spectral splitting for the BOC signal to optimize the use of frequency resources and minimize interference between new and existing signals. Improved spectral splitting capabilities can also enhance the flexibility in designing frequency-hopping patterns for FH-BOC, further bolstering anti-interference performance. In this study, we introduce a novel modulation technique known as independent frequency-hopping binary offset carrier modulation (IFH-BOC), which generalizes both BOC and FH-BOC modulations. IFH-BOC not only facilitates the asymmetric allocation of signal energy across frequencies but also allows the upper and lower sidebands of FH-BOC to operate according to distinct frequency-hopping patterns, thereby enhancing the flexibility of signal design and anti-interference performance.

We begin by establishing a mathematical model and outlining potential applications for IFH-BOC. Subsequently, we explore generation and acquisition strategies. Performance simulations and analyses are conducted for specific BOC, FH-BOC, and IFH-BOC signals. Finally, we present our conclusions.

## 2. Signal Model

Considering a BPSK signal modulated by independent frequency-hopping subcarriers in the upper and lower sidebands around the RF center frequency, the modulated signal can be represented as follows:

$$s(t) = \sum_{k=-\infty}^{+\infty} c_k \times \kappa(t - kT_c) \left[ e^{-j2\pi f_L^k(t-kT_c)} + e^{j2\pi f_U^k(t-kT_c)} \right] \tag{1}$$

where $c_k$ denotes the spreading code chip modulated with data, and $\kappa(t)$ represents a rectangular pulse of duration $T_c$; $f_L^k$ and $f_U^k$ represent the frequencies of the lower and upper sidebands of $s(t)$, respectively. By letting $f_I^k = f_U^k - f_L^k$, (1) can be expressed as:

$$\begin{aligned} s(t) &= \sum_{k=-\infty}^{+\infty} c_k \times \kappa(t - kT_c) \left[ e^{-j2\pi f_L^k(t-kT_c)} + e^{j2\pi f_L^k(t-kT_c)} e^{j2\pi f_I^k(t-kT_c)} \right] \\ &= 2 \sum_{k=-\infty}^{+\infty} c_k \times \kappa(t - kT_c) \cos\left[ (2\pi(f_L^k + 0.5 f_I^k)(t - kT_c)) \right] e^{j\pi f_I^k(t-kT_c)} \end{aligned} \tag{2}$$

Based on (2), the square of the envelope of the signal can be derived as:

$$\begin{aligned} |s(t)|^2 &= 4 \sum_{k=-\infty}^{+\infty} |c_k|^2 \times |\kappa(t - kT_c)|^2 \left| \cos\left[ (2\pi(f_L^k + 0.5 f_I^k)(t - kT_c)) \right] \right|^2 \left| e^{j\pi f_I^k(t-kT_c)} \right|^2 \\ &= 4 \sum_{k=-\infty}^{+\infty} |\kappa(t - kT_c)|^2 \left| \cos\left[ (2\pi(f_L^k + 0.5 f_I^k)(t - kT_c)) \right] \right|^2 \end{aligned} \tag{3}$$

This results in a non-constant envelope, as indicated in (3). It is necessary to reconstruct the signal represented in (2) to maintain a constant envelope. We can define the modified signal as:

$$\widetilde{s}(t) = \sum_{k=-\infty}^{+\infty} c_k \times q\left( t - kT_c, f_L^k, f_I^k \right) \tag{4}$$

and $q(t, f_1, f_2)$ is defined as

$$q(t, f_1, f_2) = \begin{cases} \text{sgn}(\cos((2\pi(f_1 + 0.5 f_2)t))) e^{j\pi f_2 t}, & 0 \le t < T_c \\ 0, & \text{otherwise} \end{cases} \tag{5}$$

where sgn refers to the sign function [23].

The square of the envelope of $\widetilde{s}(t)$ can be derived as

$$|\widetilde{s}(t)|^2 = \sum_{k=-\infty}^{+\infty} |c_k|^2 \left| q\left( t - kT_c, f_L^k, f_I^k \right) \right|^2 = 1 \tag{6}$$

$\widetilde{s}(t)$ is a constant envelope signal, which is the signal model of IFH-BOC.

The IFH-BOC signal can be denoted as IFH-BOC[$\alpha$, $\beta$]:

$$\begin{cases} \boldsymbol{\alpha} = \begin{bmatrix} \boldsymbol{\alpha}^L \\ \boldsymbol{\alpha}^U \end{bmatrix} = \begin{bmatrix} \alpha_{N-1}^L & \alpha_d^L & \alpha_0^L \\ \alpha_{M-1}^U & \alpha_d^U & \alpha_0^U \end{bmatrix} \\ \boldsymbol{\beta} = \beta \times \mathbf{I}_{2 \times 1} \end{cases} \tag{7}$$

where $\alpha_{j-1}^i = \max\{f_i^k\}/f_0$, $\alpha_0^i = \min\{f_i^k\}/f_0$, for all $i \in [L, U]$, $j \in [N, M]$, where $f_0$ is the reference frequency, equal to 1.023 MHz by default; $N$ and $M$ represent the number of frequencies of the lower and upper sidebands, respectively; $\alpha_d^i = f_d^i/f_0$, $i \in [L, U]$, with $f_d^i$ denoting the minimum frequency-hopping interval; $\beta = f_c/f_0$, with $f_c$ indicating the rate of the spreading code chip. The frequency-hopping rate is denoted as $f_v$, which is assumed to be equal to $f_c$ but is also allowed to be slower than $f_c$. Figure 1 displays both the

real and imaginary components of an example signal, IFH-BOC ([14, 1, 8; 14, 1, 1], [1, 1]), demonstrating how the signal waves vary as the frequency-hopping patterns shift.

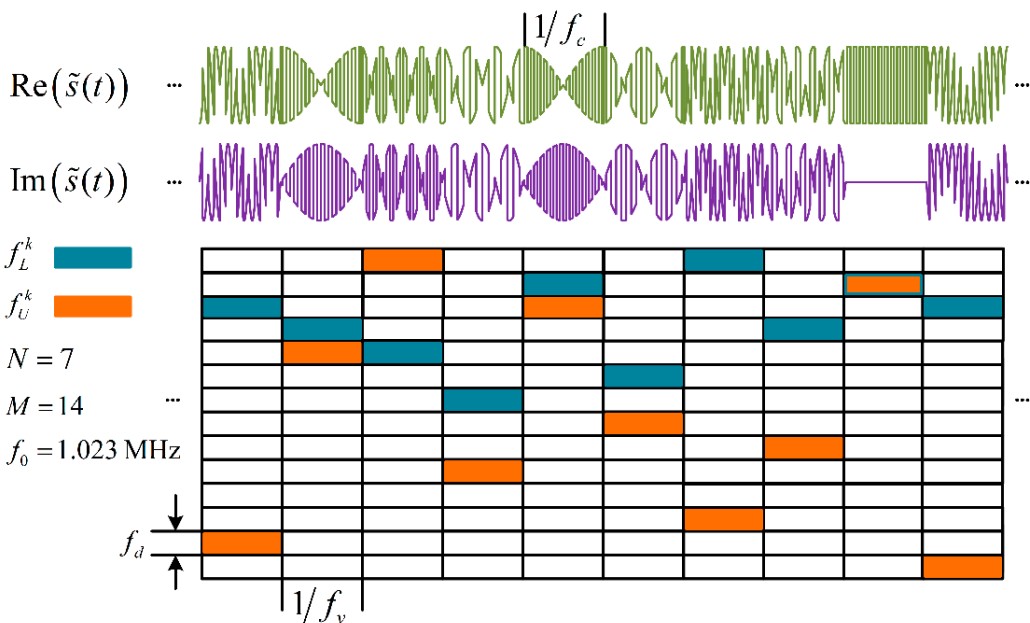

**Figure 1.** The real and the imaginary components of IFH-BOC ([14, 1, 8; 14, 1, 1], [1, 1]).

The ACF and PSD define the time–frequency characteristics of GNSS signals. The ACF and PSD can be derived for this modulation under the assumption of an ideal spreading code for IFH-BOC. The normalized PSD for IFH-BOC can be found in Appendix A, where it is presented as follows:

$$\widetilde{G}(f) = \frac{1}{I_c T_c} \sum_{i=0}^{N-1} \sum_{j=0}^{M-1} I_{i,j} \left( \frac{\left(1 - \cos\left(\pi\left(f - 0.5 f_I^{i,j}\right) t_s^{i,j}\right)\right) \sin\left(\pi\left(f - 0.5 f_I^{i,j}\right) T_c\right)}{\pi f \cos\left(\pi\left(f - 0.5 f_I^{i,j}\right) t_s^{i,j}\right)} \right)^2 \tag{8}$$

where $t_s^{i,j} = 0.5/\left(f_L^i + 0.5 f_I^{i,j}\right)$; $f_L^i$, $i = 0, 1, 2, \ldots, N-1$, denote $N$ possible single subcarrier frequencies belonging to the hopset of $\left\{f_L^k\right\}$. $f_I^{i,j} = f_U^j - f_L^i$, $j = 0, 1, 2, \ldots, M-1$; and $f_U^j$ denotes $M$ possible single subcarrier frequencies belonging to the hopset of $\left\{f_U^k\right\}$. Furthermore, the variable $I_c$ signifies the total count of occurrences across all frequency channels, while $I_{i,j}$ denotes the count of occurrences for the specific channel $(f_L^i, f_U^j)$.

## 3. Different Types of IFH-BOC

In contrast with BOC and FH-BOC modulations, IFH-BOC allows for the independent allocation of frequencies in the upper and lower sidebands of the signal. This advancement enhances the design flexibility and the anti-interception and anti-interference capabilities. The next subsection outlines several standard design schemes for IFH-BOC parameters.

In some scenarios, it is expected to further improve the anti-interference and anti-interception performance via signal design, while the signal remains constant-envelope. The signal can be denoted as IFH-BOC[α, β]:

$$\begin{cases} \boldsymbol{\alpha} = \begin{bmatrix} \alpha_{N-1}^L & 1 & 1 \\ \alpha_{M-1}^U & 1 & 1 \end{bmatrix} \\ \boldsymbol{\beta} = \mathbf{I}_{2\times 1} \end{cases} \tag{9}$$

For convenience, this IFH-BOC signal is abbreviated as Type-I IFH-BOC. The frequency-hopping patterns of the upper and lower sidebands in Type-I FH-BOC operate indepen-

dently. In other words, the switching law of the frequencies of the upper and lower sidebands is different, which increases the difficulty of jamming and eavesdropping signals by the jammer. Figures 2 and 3 show the ACF and PSD corresponding to IFH-BOC ([14, 1, 1; 14, 1, 1], [1, 1]), respectively. The shapes of the ACF and PSD are the same as FH-BOC (14, 1, 1, 1), indicating that the frequency-hopping pattern does not affect the time–frequency characteristics for the IFH–BOC signal.

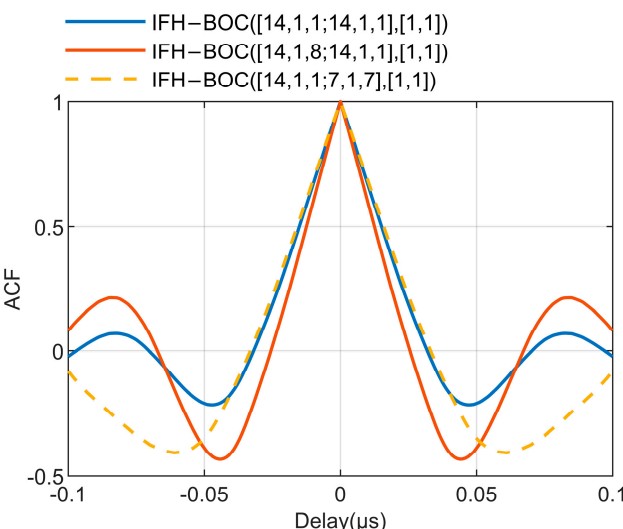

**Figure 2.** ACFs corresponding to the example signals of Type-I, Type-II, and Type-III IFH-BOC.

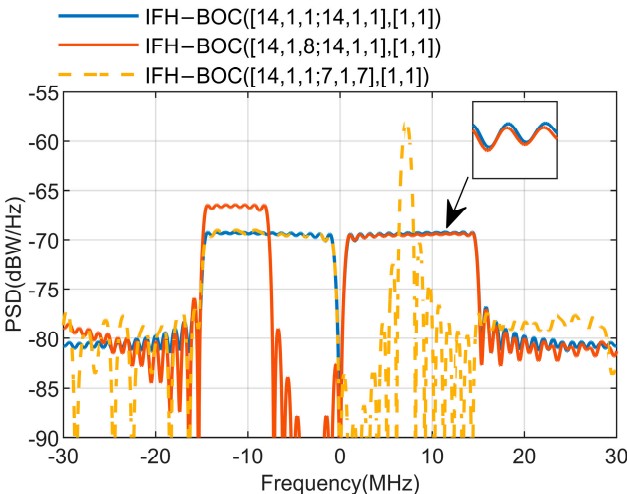

**Figure 3.** PSDs corresponding to the example signals Type-I, Type-II, and Type-III IFH-BOC.

With the development of GNSS, there are several signals to transmit to meet different navigation and positioning services. Hence, the interference between the new and the heritage signals has become one of the main factors considered in navigation signal design. A new navigation signal should have excellent spectral separation ability from the existing signals in some applications. Type-II IFH-BOC is proposed for this purpose, denoted as IFH-BOC[$\alpha$, $\beta$]:

$$\begin{cases} \boldsymbol{\alpha} = \begin{bmatrix} \alpha_{N-1}^{L} & 1 & \alpha_{0}^{L} \\ \alpha_{M-1}^{U} & 1 & \alpha_{0}^{U} \end{bmatrix} \\ \boldsymbol{\beta} = \mathbf{I}_{2\times1} \end{cases} \tag{10}$$

Figures 2 and 3 illustrate the ACF and PSD corresponding to IFH-BOC ([14, 1, 8; 14, 1, 1], [1, 1]), respectively. The signal power across the upper and lower sidebands of the signal can be independently allocated by varying the modulation parameters of these bands.

Although FH-BOC signals have better anti-interference performance and lower ACF side peaks than BOC signals, the processing complexity of FH-BOC signals is higher than BOC signals. We propose the Type-III IFH-BOC signal to overcome this problem, denoted as IFH-BOC[$\alpha$, $\beta$]:

$$\begin{cases} \boldsymbol{\alpha} = \begin{bmatrix} \alpha_{N-1}^L & 1 & \alpha_0^L \\ \alpha_0^U & 1 & \alpha_0^U \end{bmatrix} \\ \boldsymbol{\beta} = \mathbf{I}_{2 \times 1} \end{cases} \tag{11}$$

Figures 2 and 3 illustrate the ACF and PSD corresponding to IFH-BOC ([14, 1, 1; 7, 1, 7], [1, 1]), respectively. The frequencies of the lower sideband in the IFH-BOC signal vary randomly according to the frequency-hopping pattern, whereas the upper sideband functions as a BPSK signal. If minimizing signal processing complexity is a priority, the receiver may focus solely on the upper sideband of the IFH-BOC signal. Conversely, the receiver can enhance its resistance to interference by processing both the upper and lower sidebands simultaneously.

## 4. Generation and Acquisition Scheme

IFH-BOC represents a generalized form of both FH-BOC and BOC, and its generation method can be achieved by adapting the existing schemes of these two modulations. Figure 4 illustrates the architecture of the IFH-BOC signal. The corresponding hopping frequencies are determined based on the frequency-hopping patterns for both the lower and upper sidebands. Subsequently, the generators produce square, cosine, and sine waves. The data message, spreading code, and square wave are then multiplied with the cosine and sine waves to yield the in-phase component $I(t)$ and the quadrature component $Q(t)$. Finally, these components, modulated onto cosine-phase and sine-phase radio frequency carriers, are combined to form the complete signal. A Type-II IFH-BOC signal, named IFH-BOC ([14, 1, 8; 14, 1, 1], [1, 1]), is generated using this method. The simulated and theoretical ACF and PSD curves are displayed in Figure 5.

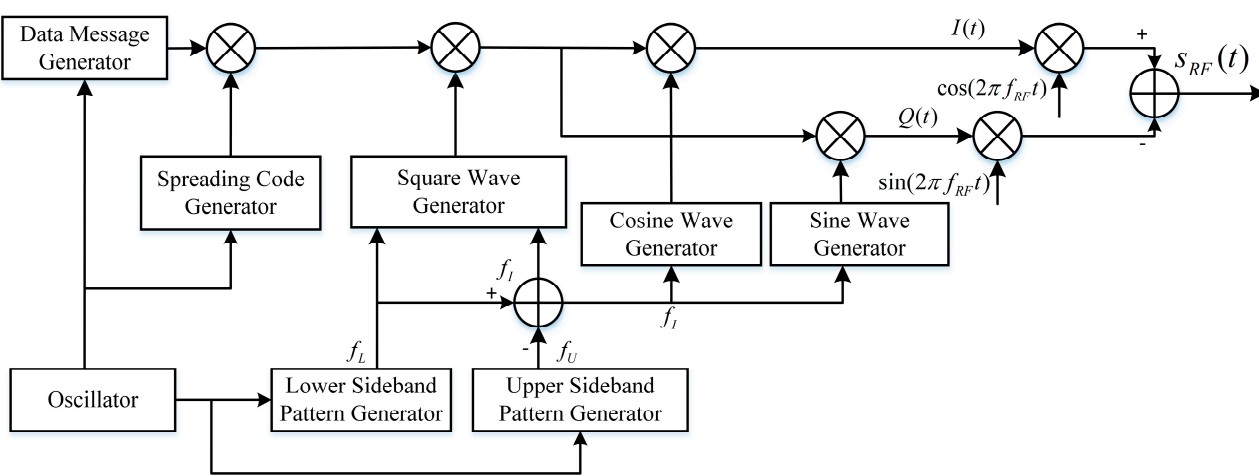

**Figure 4.** The architecture of the IFH-BOC signal generation.

The acquisition process for the FH-BOC signal involves searching across three dimensions: the code dimension, frequency-hopping, and Doppler dimensions [19]. In contrast, the IFH-BOC signal introduces a fourth dimension by splitting the frequency-hopping dimension into two: one for the upper sideband and one for the lower sideband. Consequently, the search for the IFH-BOC signal operates in four dimensions. Similar to the FH-BOC signal, the average acquisition time for the IFH-BOC signal can be estimated as follows:

$$T_{acq} = \frac{1}{2} \frac{f_{unc} t_{unc} N_T^L N_T^U T_{dwell}}{f_{bin} t_{bin}} \tag{12}$$

In this equation, the variable $f_{unc}$ represents the Doppler frequency uncertainties for the IFH-BOC signal, while $t_{unc}$ denotes the uncertainty in the phase of the spreading code; the term $f_{bin}$ corresponds to the Doppler frequency search step, and $t_{bin}$ corresponds to the spreading code phase search step; the variable $N_T^L$ and $N_T^U$ represent the ratios of the periods of the frequency-hopping patterns for the upper and lower sidebands to the spreading code, respectively; and $T_{dwell}$ denotes the search dwell time. The frequency-hopping patterns significantly impact both the acquisition time and complexity associated with the IFH-BOC signal. One can set the period of the frequency-hopping pattern to match that of the spreading code to streamline the search process, allowing the phases of the frequency-hopping patterns to be derived from the spreading code phase. In this scenario, the acquisition time and complexity for the IFH-BOC signal would align with those of the BOC signal.

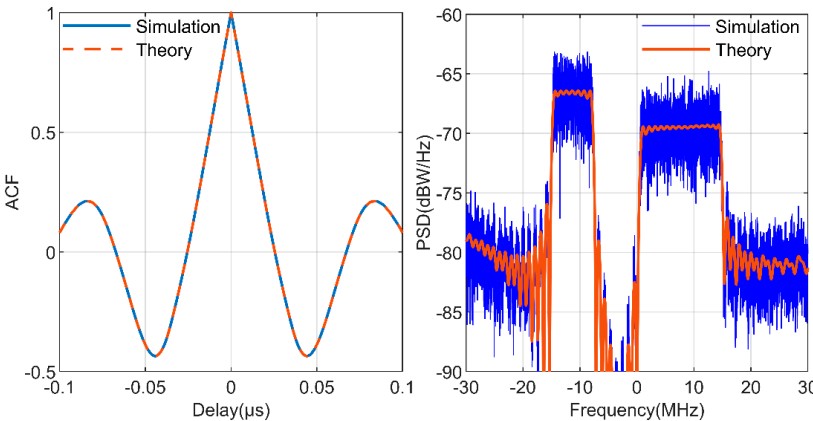

**Figure 5.** Theoretical and simulated curves of the ACF and PSD for IFH-BOC ([14, 1, 8; 14, 1, 1], [1, 1]).

Similar to FH-BOC and BOC signals, IFH-BOC can be processed using two primary methods: processing each sideband separately or utilizing a wideband receiver to analyze the entire IFH-BOC signal for enhanced performance. A receiver designed for FH-BOC or BOC can handle the IFH-BOC signal with minimal adjustments. Figure 6 illustrates a parallel acquisition scheme for the spreading code phase and the frequency-hopping patterns of IFH-BOC. Initially, the intermediate frequency (IF) signal is down-converted using the local carrier and then individually subjected to Fourier transformation. Next, the local spreading code, square wave, and sine, and cosine waves are multiplied, followed by a Fourier transform (FFT) on the resulting data, which are then conjugated. The results from the first two steps are multiplied together, and an inverse Fourier transform (IFFT) is performed. Ultimately, the phase and frequency associated with the maximum modulus of the IFFT results indicate the acquisition outcomes.

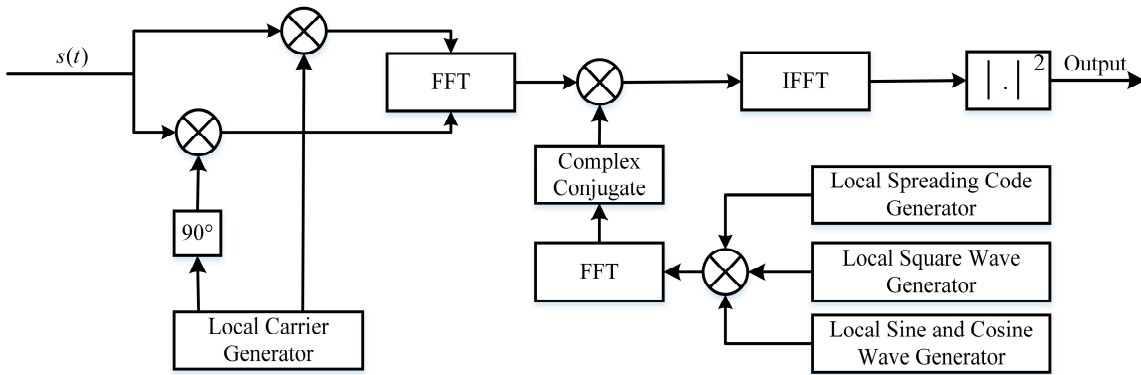

**Figure 6.** The architecture of IFH-BOC signal acquisition.

## 5. Performance Analysis

This section examines the performance of IFH-BOC modulation relative to FH-BOC and BOC modulations. The selected signals, labeled S1–S6, are detailed in Table 1. S1 represents an FH-BOC signal, while S6 refers to BOC(10,5), the military signal used on GPS carriers L1 and L2 [3]. S2–S5 comprise various typical IFH-BOC signals. Figure 7 illustrates the frequency-hopping patterns of these selected signals, revealing that S1 and S6 share the same patterns for their lower and upper sidebands, whereas S2–S5 exhibit distinct patterns.

**Table 1.** Parameters of the selected signals.

| IFH-BOC | $\alpha_{N-1}^L$ | $\alpha_d^L$ | $\alpha_0^L$ | $\alpha_{M-1}^U$ | $\alpha_d^U$ | $\alpha_0^U$ | $\beta$ | $N$ | $M$ | Type | $\{f_L^k\}=\{f_U^k\}$ |
|---------|------------------|--------------|--------------|------------------|--------------|--------------|---------|-----|-----|------|------------------------|
| S1 | 14 | 1 | 1 | 14 | 1 | 1 | 1 | 14 | 14 | FH-BOC | Yes |
| S2 | 14 | 1 | 1 | 14 | 1 | 1 | 1 | 14 | 14 | Type-I | No |
| S3 | 14 | 1 | 1 | 5 | 1 | 5 | 1 | 14 | 1 | Type-III | No |
| S4 | 14 | 1 | 8 | 10 | 1 | 1 | 1 | 7 | 10 | Type-II | No |
| S5 | 14 | 1 | 8 | 14 | 1 | 8 | 1 | 7 | 7 | Type-II | No |
| S6 | 10 | 1 | 10 | 10 | 1 | 10 | 5 | 1 | 1 | BOC | Yes |

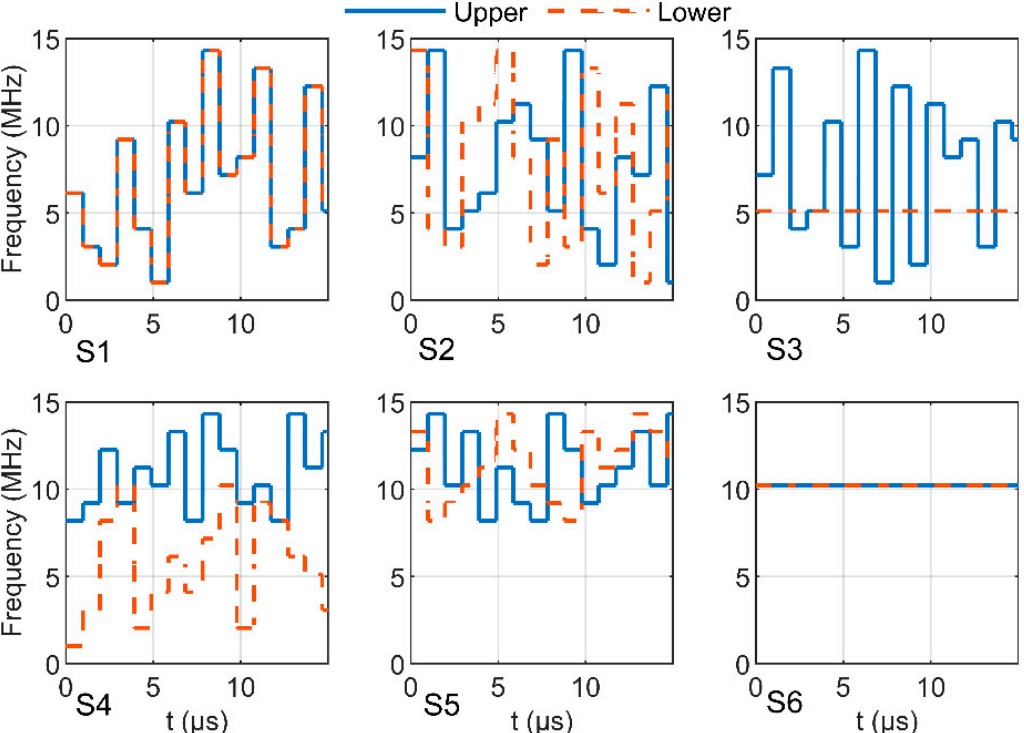

**Figure 7.** Frequency-hopping patterns for the selected signals.

### 5.1. Comparison of ACFs and PSDs

The ACFs for the selected signals are illustrated in Figure 8. Initially, the ACF of signal S1 aligns with that of S2, while the main lobe of S5 closely matches that of S6. Furthermore, the zero crossings nearest to the main peak (ZCNMs) for S5 and S6 are the smallest among the selected signals, whereas S3 exhibits the largest ZCNMs. The ZCNMs for S1 and S2 fall between those of S3 and S4, being smaller than S3 but larger than S4. Notably, the maximum side peak to main peak ratio (MSR) of S6 is the highest among the selected signals, while S1 and S2 have the lowest MSR. The MSR for S4 is greater than that of S1 and S2, but less than that of S3.

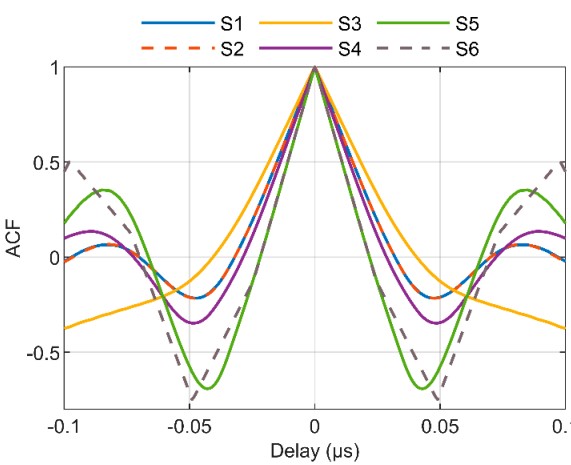

**Figure 8.** Normalized ACFs for the selected signals.

Figure 9 displays the PSDs for the selected signals. It is evident that the PSD envelopes for S1 and S2 are identical, and the lower sideband of S3 aligns with those of S1 and S2, while its upper sideband coincides with that of BOC(7,1). Among the selected signals, S3 has the highest maximum value of the PSD (MVP), whereas S1 and S2 have the lowest MVPs. The MVP of the upper sideband of S4 is less than that of its lower sideband, and the lower sideband MVP of S3 is also lower than its upper sideband.

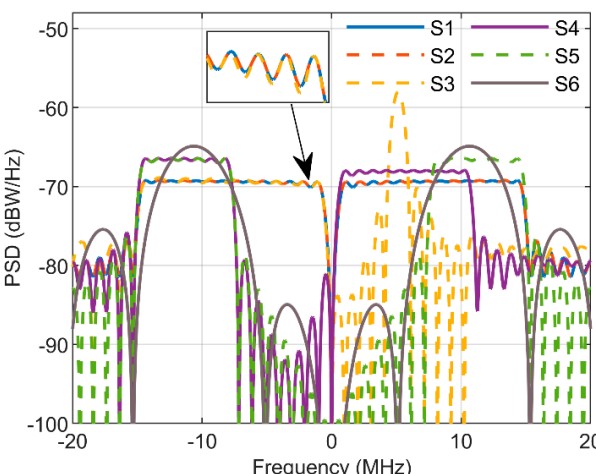

**Figure 9.** Normalized PSDs for the selected signals.

The comparison and analysis of the ACFs and PSDs for the selected signals reveal that IFH-BOC modulation retains the time–frequency characteristics of both FH-BOC and BOC modulations while offering greater flexibility in frequency allocation. This advantage facilitates more efficient use of frequency resources, minimizes interference with other signals, and enhances anti-interference capabilities. IFH-BOC represents a generalized modulation encompassing both FH-BOC and BOC techniques.

### 5.2. Code Tracking Accuracy

The signal modulation scheme primarily dictates the maximum accuracy of code tracking. In this subsection, we will evaluate the tracking performance of the selected signals.

### 5.2.1. Root Mean Square (RMS) Bandwidth

The RMS bandwidth $\beta_{\text{rms}}$ of a band-limited signal is defined as follows

$$\beta_{\text{rms}} = \sqrt{\int_{-\beta_r/2}^{\beta_r/2} f^2 \overline{G}_s(f) df} \tag{13}$$

where $\beta_r$ represents the front-end bandwidth of the receiver, $G_s(f)$ denotes the PSD of the band-limited signal, and $\overline{G}_s(f) = G_s(f) / \int_{-\beta_r/2}^{\beta_r/2} G_s(f) df$ signifies the normalized PSD [24]. An increase in $\beta_{\text{rms}}$ correlates with a decrease in the bound of the code tracking error. Figure 10 illustrates $\beta_{\text{rms}}$ for various example signals. When $\beta_r$ is set at 30 MHz, signal S5 exhibits the largest $\beta_{\text{rms}}$ among the examples, whereas S3 displays the smallest. Both S1 and S2 share the same $\beta_{\text{rms}}$, while S4's $\beta_{\text{rms}}$ is less than that of S6 but greater than those of S1 and S2.

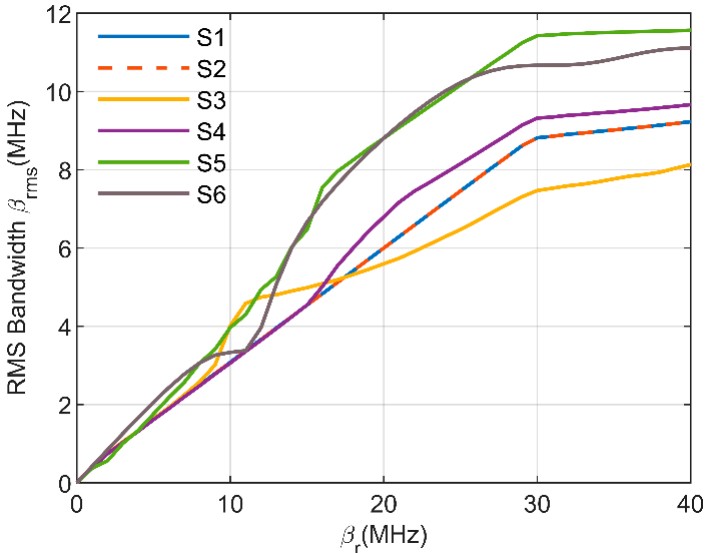

**Figure 10.** RMS bandwidths for the selected signals.

### 5.2.2. Lower Bound of Code Tracking Error

The lower bound of the code tracking error indicates the maximum achievable tracking accuracy, expressed as

$$\sigma_{\text{LB}} = \frac{1}{2\pi\beta_{\text{rms}}} \sqrt{\frac{B_L(1 - 0.5B_L T)}{C/N_0 \int_{-\beta_r/2}^{\beta_r/2} \overline{G}_s(f) df}} \tag{14}$$

Here, $B_L$ represents the equivalent rectangular bandwidth of the tracking loop, $C/N_0$ denotes the carrier-to-noise density ratio, and $T$ indicates the integration time [24]. The integral terms in (13) and (14) indicate that the high-frequency components of the signal have a significant impact on the code tracking accuracy. Increasing the high-frequency power of the signal can improve the code tracking accuracy. The code tracking accuracy for IFH-BOC signals can be improved by increasing $\alpha_{j-1}^i$ or $\alpha_0^i$, where $i \in [L, U]$, and $j \in [N, M]$. Increasing $\beta$ can also achieve this, but requires a wider signal bandwidth. Table 1 lists $\alpha_{j-1}^i$, $\alpha_0^i$, and $\beta$ for the selected signals S1~S6. For a $B_L$ of 1 Hz, a $T$ of 1 ms, and a $\beta_r$ of 40 MHz, Figure 11 illustrates the lower bounds of the code tracking errors for the selected signals. Among these, S5 exhibits the smallest lower bound, while S3 has the largest. The lower bounds for S1 and S2 are marginally less than that of S3, and the lower bound for S4 is less than those for S1 and S2, yet greater than that of S6.

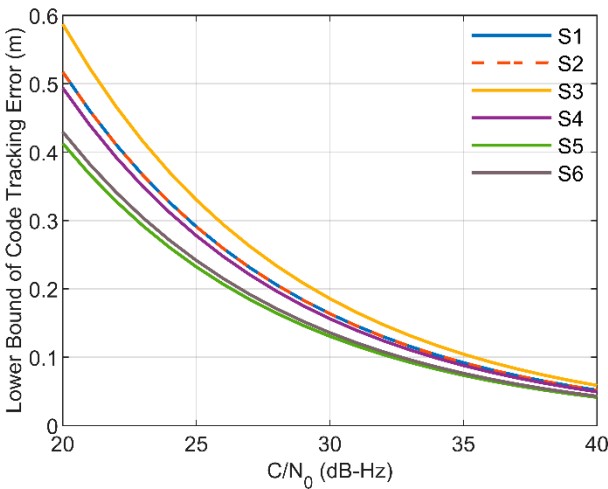

**Figure 11.** Lower bounds of the code tracking errors for the selected signals.

### 5.3. Performance of Noncoherent Early–Late Processing (NELP)

When white noise and interference are present, and the tracking error of the code loop is sufficiently small to allow for a linear analysis of its operation, the variance of the code tracking error for NELP can be represented as [25]:

$$
\sigma^2_{\text{NELP}} = \frac{B_L(1 - 0.5B_LT)\int_{-\beta_r/2}^{\beta_r/2}\left[1 + \frac{C_I}{N_0}G_I(f)\right]G_s(f)\sin^2(\pi f\Delta)df}{\frac{C}{N_0}\left(2\pi\int_{-\beta_r/2}^{\beta_r/2}fG_s(f)\sin^2(\pi f\Delta)df\right)^2}
$$
$$
\times\left[1 + \frac{\int_{-\beta_r/2}^{\beta_r/2}G_s(f)\cos^2(\pi f\Delta)df}{T\frac{C}{N_0}\left(\int_{-\beta_r/2}^{\beta_r/2}G_s(f)\cos(\pi f\Delta)df\right)^2} + \frac{\int_{-\beta_r/2}^{\beta_r/2}G_I(f)G_s(f)\cos^2(\pi f\Delta)df}{T\frac{C}{C_I}\left(\int_{-\beta_r/2}^{\beta_r/2}G_s(f)\cos(\pi f\Delta)df\right)^2}\right] \quad (15)
$$

In this formula, $C_I/N_0$ represents the ratio of the interference carrier power to noise density, $G_I(f)$ denotes the PSD of the interference signal, and $\Delta$ indicates the early-late spacing. With zero interference carrier power, a $\beta_r$ of 40 MHz, a $T$ of 0.02 s, a $\Delta$ of 32 ns, and a $B_L$ of the code tracking loop set to 1 Hz, Figure 12 compares the NELP code tracking errors for the selected signals as a function of $C/N_0$. The NELP performance for these selected signals aligns with the lower bounds of the code-tracking error.

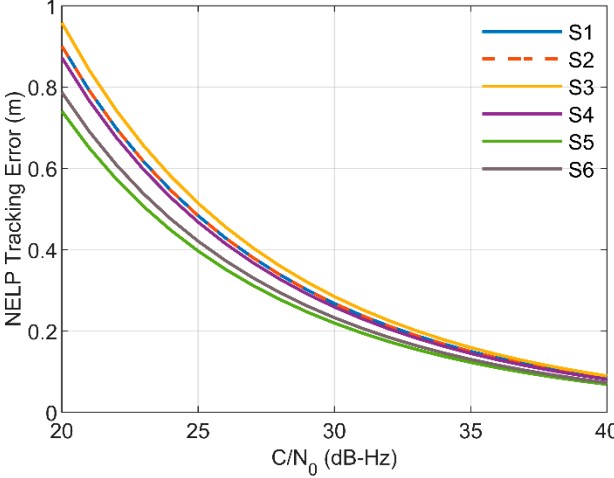

**Figure 12.** The code tracking errors for the selected signals with $\beta_r = 40$ MHz.

*5.4. Anti-Interference Performance*

This section evaluates the anti-interference performance of the selected signals against narrowband interference. The processing gain serves as a metric for assessing anti-interference performance in spread spectrum signals, with a higher processing gain indicating superior performance. The processing gains for BOC, FH-BOC, and IFH-BOC signals are derived in Appendix B. The processing gain for BOC ($\alpha$, $\beta$) is expressed as follows:

$$G_{\text{BOC}} \approx 10 \log(\beta \frac{f_0}{f_D}) + 3 \tag{16}$$

where $f_D$ represents the data rate. The processing gain for FH-BOC ($\alpha_{M-1}$:1:$\alpha_0$, $\beta$) can be expressed as

$$G_{\text{FH-BOC}} \approx 10 \log(\beta \frac{f_0}{f_D}) + 10 \log(\frac{\alpha_{M-1} - \alpha_0}{\beta} + 1) + 3 \tag{17}$$

and, for an IFH-BOC[$\boldsymbol{\alpha}$, $\boldsymbol{\beta}$],

$$\begin{cases} \boldsymbol{\alpha} = \begin{bmatrix} \boldsymbol{\alpha}^L \\ \boldsymbol{\alpha}^U \end{bmatrix} = \begin{bmatrix} \alpha_{N-1}^L & 1 & \alpha_0^L \\ \alpha_{M-1}^U & 1 & \alpha_0^U \end{bmatrix} \\ \boldsymbol{\beta} = \beta \times \mathbf{I}_{2 \times 1} \end{cases} \tag{18}$$

the processing gain can be expressed as

$$G_{\text{IFH-BOC}} \approx 10 \log(\beta \frac{f_0}{f_D}) + 10 \log(\frac{\alpha_{N-1}^L - \alpha_0^L}{\beta} + \frac{\alpha_{M-1}^U - \alpha_0^U}{\beta} + 2) \tag{19}$$

Table 2 summarizes the processing gains for the selected signals. S1 and S2 exhibit the highest processing gains, whereas S6 has the lowest. Additionally, S3's processing gain surpasses that of S5 but remains below that of S4. Generally, for both BOC and FH-BOC signals, a greater processing gain correlates with enhanced anti-interference performance.

**Table 2.** Processing gains for the selected signals.

| Parameter | S1 | S2 | S3 | S4 | S5 | S6 |
|---|---|---|---|---|---|---|
| $G$(dB) | 57.6 | 57.6 | 54.9 | 55.4 | 54.6 | 53.1 |

Narrowband interference typically refers to interference with a bandwidth of less than 10% of the overall signal bandwidth [26]. The impact of this interference on the receiver increases as its frequency approaches the central frequency of the main lobe in the GNSS signal power spectrum. The PSD for narrowband interference $s_I(t)$ is generally defined as

$$G_I(f) = \begin{cases} \frac{1}{2\beta_I}, & ||f| - f_I| \le \frac{\beta_I}{2} \\ 0, & \text{others} \end{cases} \tag{20}$$

The PSD for $s_I(t)$ exhibits two symmetrical rectangular sidebands with a bandwidth of $\beta_I$, centered at frequencies $\pm f_I$ [26]. Figure 13 shows the PSD for the narrowband interference $s_I(t)$.

For a $\beta_r$ of 40 MHz, a $C/N_0$ of 45 dB, and some narrowband interference with bandwidths of 10 KHz centered on the center frequencies of the sidebands of the selected signals, Figure 14 illustrates the effective $C/N_0$ versus $C_I/N_0$ for the selected signals. The results indicate that the effective $C/N_0$ of all signals, except for S3, aligns with the processing gain comparisons. While S3 does not possess the highest processing gain, it demonstrates superior resistance to narrowband interference among the selected signals. This is due to the differing center frequencies of its upper and lower sidebands, with the upper sideband's center frequency positioned far from the interference's center frequency.

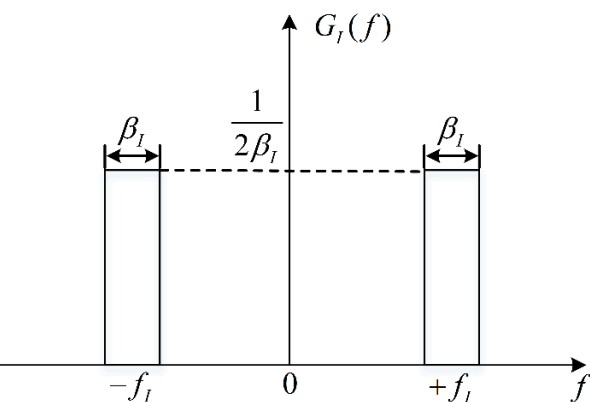

**Figure 13.** PSD for the narrowband interference.

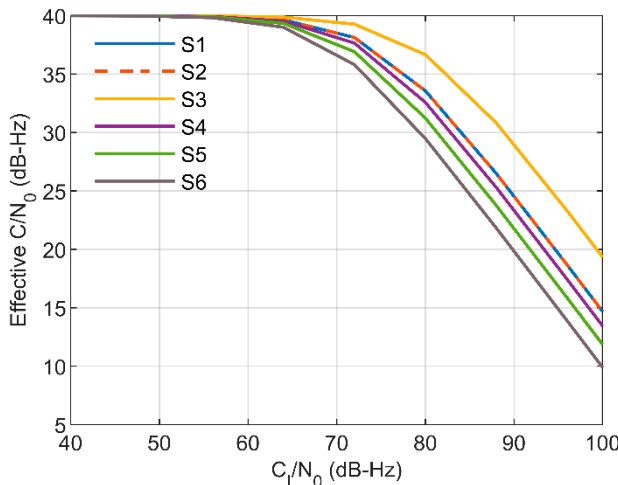

**Figure 14.** Effective $C/N_0$ for the selected signals.

### 5.5. Multipath Performance

Multipath error constitutes a significant source of inaccuracies in satellite navigation systems. Environmental reflections around the receiver can distort pseudo code and carrier phase measurements. The extent of this error is influenced by the surrounding environment of the antenna, with a small spatial correlation making it challenging to mitigate effectively [27]. Consequently, the resistance of GNSS signals to multipath interference is a crucial consideration during the signal design phase.

Given the environmental factors affecting the receiver antenna, the multipath signals entering the receiver may follow one or multiple reflection paths, complicating quantitative analysis of the multipath effect. Therefore, a common approach in navigation signal design involves using a model that analyzes a single reflection path, and the multipath error is expressed as

$$\varepsilon_\tau(\widetilde{\tau}_1) \approx \frac{\widetilde{a}_1 \cos(\varphi) \int_{-\beta_r/2}^{\beta_r/2} G_x(f) \sin(2\pi f \widetilde{\tau}_1) \sin(\pi f \Delta) df}{2\pi \int_{-\beta_r/2}^{\beta_r/2} f G_x(f) \sin(\pi f \Delta)(1 + \cos(\varphi) \cos(2\pi f \widetilde{\tau}_1)) df} \tag{21}$$

where $G_x(f)$ represents the PSD of the direct signal, $\widetilde{a}_1$ denotes the amplitude ratio of the multipath signal to the direct signal, $\widetilde{\tau}_1$ indicates the additional delay of the multipath signal relative to the direct signal, and $\varphi$ is the carrier phase of the multipath signal. The multipath error reaches its extreme values when the carrier phase of the multipath signal is $0°$ or $180°$, and the multipath error envelopes are defined as follows [23]:

$$\mathrm{E}_\tau(\widetilde{\tau}_1) = \left( \varepsilon_\tau(\widetilde{\tau}_1)|_{\varphi=0°}, \varepsilon_\tau(\widetilde{\tau}_1)|_{\varphi=180°} \right) \tag{22}$$

To further evaluate the multipath performance over the possible multipath delay range, we define the average multipath error envelope as follows

$$\Gamma_\tau(\widetilde{\tau}_1') = \frac{1}{\widetilde{\tau}'} \int_0^{\widetilde{\tau}'} \frac{\left[ \mathrm{abs}\left( \varepsilon_\tau(\widetilde{\tau}_1)|_{\varphi=0^\circ} \right) + \mathrm{abs}\left( \varepsilon_\tau(\widetilde{\tau}_1)|_{\varphi=180^\circ} \right) \right]}{2} d\widetilde{\tau}_1 \qquad (23)$$

where abs $(\cdot)$ denotes the absolute value function.

In a scenario with one direct path and one reflected path, where $\widetilde{a}_1$ is $-6$ dB, early-late spacings of 32 ns for S1 and S2, 40 ns for S3, 28 ns for S4, and 22 ns for S5 and S6 are utilized, along with a $\beta_r$ of 40 MHz. Figures 15 and 16 illustrate the multipath error envelopes against the multipath delay and the average multipath error envelopes for the selected signals, respectively. When the multipath delay is under 15 m, the error envelopes for S5 are the smallest, while S3 exhibits the largest. The error envelopes for S6 are greater than those for S1 and S2 but smaller than those for S4. The average multipath error envelopes for S1 and S2 are the lowest, whereas S6 has the highest. The average multipath error envelope for S4 closely aligns with that of S3.

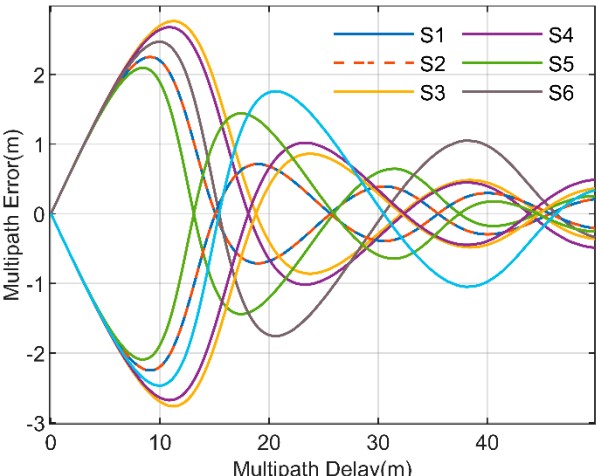

**Figure 15.** Multipath errors for the selected signals.

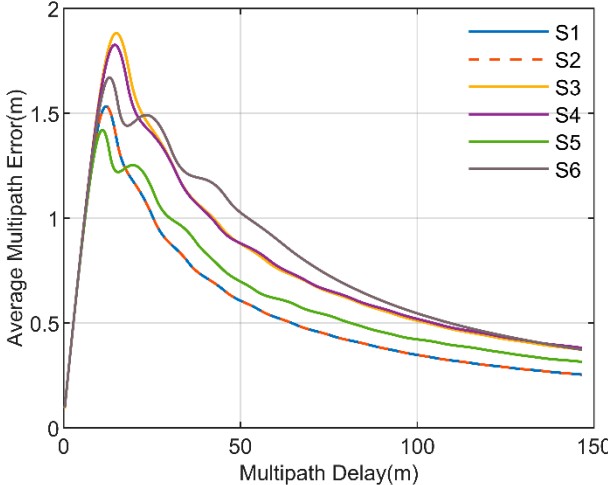

**Figure 16.** Average multipath errors for the selected signals.

## 6. Conclusions

This study introduced a novel modulation scheme called IFH-BOC, which advances the concept of FH-BOC by employing distinct frequency-hopping patterns for the upper and lower sidebands. The signal model, various design types, generation and acquisition

methods, and the characteristics of ACF and PSD are explored. While not exhaustive, it highlighted several typical applications of IFH-BOC. The generation scheme can be derived from modifications of existing FH-BOC and BOC techniques. Through carefully designing the frequency-hopping patterns, the acquisition time and complexity of IFH-BOC can match those of FH-BOC or BOC signals. The results of the performance analysis reveal that S1, as a Type-I IFH-BOC signal, and S2, as an FH-BOC signal, exhibit the same time–frequency characteristics. S1 and S2 exhibit the smallest ACF side lobes among the selected signals, indicating the lowest ACF peak ambiguity. Among the selected signals, the S5, as a Type-II IFH-BOC signal, demonstrates the highest code tracking accuracy under the condition of the same front-end bandwidth of the receiver. The processing gains of S1 and S2 are the highest among the selected signals. The multipath error envelopes for S5 are the smallest among the selected signals when the multipath delay is less than 15 m, and the average multipath error envelopes for S1 and S2 are the smallest among the selected signals.

The upper and lower sideband frequencies of FH-BOC signal are symmetrical hopping, while the upper and lower sideband frequencies of IFH-BOC signal are independent hopping, which increases the difficulty of being interfered, thus improving the anti-interference ability. The Type-I IFH-BOC signal is recommended for better anti-interference performance. Additionally, IFH-BOC allows energy to be allocated to the upper and lower sidebands with asymmetric frequencies, providing superior spectral splitting capability compared with BOC modulation. The Type-II IFH-BOC signal is recommended for better spectral separation ability and higher measurement accuracy. The Type-III IFH-BOC signal is recommended to reduce processing complexity due to its compatibility with the BPSK signal. IFH-BOC retains the advantageous features of FH-BOC, providing the same time–frequency characteristics and measurement accuracy under equivalent modulation parameters while offering enhanced anti-interference capabilities due to the distinct frequency-hopping patterns. The findings of this study can be applied to future GNSS signal design, including military applications and the utilization of unused L-band frequencies.

**Author Contributions:** Y.Y. and J.M. developed the conceptual framework and carried out the implementation of the IFH-BOC modulation, as well as the writing of the manuscript, while also contributing to the review and editing process. L.D. and H.L. offered theoretical insights and recommendations for the paper's revisions. All authors have read and agreed to the published version of the manuscript.

**Funding:** Funding was supported by Science and Technology Innovation 2030 Key Project of 'New Generation Artificial Intelligence' (Grant Nos.2020AAA0108200).

**Data Availability Statement:** The original contributions presented in this study are included in the article. Further inquiries can be directed to the corresponding author.

**Conflicts of Interest:** The authors declare no conflicts of interest.

## Appendix A

Assuming an ideal spreading code for the IFH-BOC, the ACF of the IFH-BOC signal can be represented as follows:

$$
\begin{aligned}
\widetilde{R}(t, t+\tau) &= E(\widetilde{s}(t)\widetilde{s}^*(t+\tau)) \\
&= E\left( \sum_{k=-\infty}^{+\infty} c_k q\left(t - kT_c, f_L^k, f_I^k\right) \sum_{l=-\infty}^{+\infty} c_l q^*\left(t + \tau - lT_c, f_L^l, f_I^l\right) \right) \\
&= \sum_{k=-\infty}^{+\infty} \sum_{l=-\infty}^{+\infty} R_c(l) E\left( q\left(t - kT_c, f_L^k, f_I^k\right) q^*\left(t + \tau - kT_c - lT_c, f_L^{k+l}, f_I^{k+l}\right) \right)
\end{aligned}
\tag{A1}
$$

In this equation, $R_c(l) = E(c_k c^*_{k+l})$ indicates the ACF of the spreading code. For an ideal spreading code, $R_c(l)$ can be expressed as follows:

$$
R_c(l) = \begin{cases} 1, & l = 0 \\ 0, & l \neq 0 \end{cases}
\tag{A2}
$$

substituting (A2) into (A1) leads to:

$$\widetilde{R}(t, t + \tau) = \sum_{k=-\infty}^{+\infty} E\left(q\left(t - kT_c, f_L^k, f_I^k\right) q^*\left(t + \tau - kT_c, f_L^k, f_I^k\right)\right) \tag{A3}$$

For the IFH-BOC signal, the probability of utilizing a specific set of upper and lower sideband frequency channels can be articulated as follows:

$$\begin{aligned}
P\left(f_L^k = f_L^i, f_U^k = f_U^j\right) &= P\left(f_I^k = f_I^{i,j} \middle| f_L^k = f_L^i\right) \\
&= \frac{I_{i,j}}{T_c}, i = 0, 1, 2, \ldots, N-1; j = 0, 1, 2, \ldots, M-1;
\end{aligned} \tag{A4}$$

Here, $I_c$ represents the total occurrences of all frequency channels, while $I_{i,j}$ signifies the number of occurrences for the channel $(f_L^i, f_U^j)$. Consequently, substituting (A4) into (A3) allows for the simplification of $\widetilde{R}(t, t + \tau)$ to:

$$\widetilde{R}(t, t + \tau) = \frac{1}{I_c} \sum_{k=-\infty}^{+\infty} \sum_{i=0}^{N-1} \sum_{j=0}^{M-1} q\left(t - kT_c, f_L^i, f_I^{i,j}\right) q^*\left(t + \tau - kT_c, f_L^i, f_I^{i,j}\right) \times I_{i,j} \tag{A5}$$

Importantly, the IFH-BOC signal is categorized as cyclostationary. The ACF, which depends solely on the variable $\tau$, can be derived by averaging the ACF over the interval $t \in [0, T_c]$:

$$\widetilde{R}(\tau) = \frac{1}{T_c} \int_0^{T_c} \widetilde{R}(t, t + \tau) dt \tag{A6}$$

Substituting (A5) into (A7), the ACF for the IFH-BOC signal can be formulated as follows:

$$\widetilde{R}(\tau) = \frac{1}{I_c T_c} \int_0^{T_c} \sum_{k=-\infty}^{+\infty} \sum_{i=0}^{N-1} \sum_{j=0}^{M-1} q\left(t - kT_c, f_L^i, f_I^{i,j}\right) q^*\left(t + \tau - kT_c, f_L^i, f_I^{i,j}\right) \times I_{i,j} dt \tag{A7}$$

Let $t' = t - kT_c$ be defined as a specific variable. Substituting this into (A8) transforms the expression into:

$$\begin{aligned}
\widetilde{R}(\tau) &= \frac{1}{I_c T_c} \sum_{k=-\infty}^{+\infty} \int_{-kT_c}^{(1-k)T_c} \sum_{i=0}^{N-1} \sum_{j=0}^{M-1} q\left(t - kT_c, f_L^i, f_I^{i,j}\right) q^*\left(t + \tau - kT_c, f_L^i, f_I^{i,j}\right) \times I_{i,j} dt \\
&= \frac{1}{I_c T_c} \int_{-\infty}^{+\infty} \sum_{i=0}^{N-1} \sum_{j=0}^{M-1} q\left(t', f_L^i, f_I^{i,j}\right) q^*\left(t' + \tau, f_L^i, f_I^{i,j}\right) \times I_{i,j} dt \\
&= \frac{1}{I_c T_c} \sum_{i=0}^{N-1} \sum_{j=0}^{M-1} I_{i,j} R_q^{i,j}(\tau)
\end{aligned} \tag{A8}$$

The PSD of $\widetilde{s}(t)$ is obtained via the Fourier transform of its ACF:

$$\widetilde{G}(f) = \text{FT}\left[\widetilde{R}(\tau)\right] \tag{A9}$$

Therefore, substituting (A9) into (A10) enables the derivation of the PSD for the IFH-BOC signal as follows:

$$\widetilde{G}(f) = \frac{1}{I_c T_c} \sum_{i=0}^{N-1} \sum_{j=0}^{M-1} I_{i,j} G_q^{i,j}(f) \tag{A10}$$

where $G_q^{i,j}(f)$ denotes the PSD of $q\left(t, f_L^i, f_I^{i,j}\right)$, derived as

$$G_q^{i,j}(f) = \left( \frac{\left(1 - \cos\left(\pi\left(f - 0.5f_I^{i,j}\right)t_s^{i,j}\right)\right)\sin\left(\pi\left(f - 0.5f_I^{i,j}\right)T_c\right)}{\pi f \cos\left(\pi\left(f - 0.5f_I^{i,j}\right)t_s^{i,j}\right)} \right)^2 \tag{A11}$$

By incorporating (A11) into (A10), the PSD of the IFH-BOC signal can ultimately be articulated as:

$$\widetilde{G}(f) = \frac{1}{I_c T_c} \sum_{i=0}^{N-1} \sum_{j=0}^{M-1} I_{i,j} \left( \frac{\left(1 - \cos\left(\pi\left(f - 0.5f_I^{i,j}\right)t_s^{i,j}\right)\right)\sin\left(\pi\left(f - 0.5f_I^{i,j}\right)T_c\right)}{\pi f \cos\left(\pi\left(f - 0.5f_I^{i,j}\right)t_s^{i,j}\right)} \right)^2 \tag{A12}$$

**Appendix B**

In a direct-sequence spread spectrum system, the factor by which the bandwidth of the data signal is expanded due to spread spectrum modulation is known as the processing gain [23]. This gain is defined as the ratio of the spreading code chip rate to the data rate:

$$G_{\text{DS}} \approx 10\log\left(\frac{f_c}{f_D}\right) \tag{A13}$$

Based on (A13), the processing gain of BOC $(\alpha, \beta)$ can be calculated as follows:

$$G_{\text{BOC}} \approx 10\log\left(\beta\frac{f_0}{f_D}\right) + 3 \tag{A14}$$

where the processing gain is increased by 3 dB due to the spectral splitting characteristic of the BOC signal.

In addition to the processing gain provided by direct-sequence spread spectrum modulation, both FH-BOC and IFH-BOC also benefit from frequency-hopping spread spectrum gain. Their frequency-hopping spread spectrum gain is defined as follows:

$$G_{\text{FS}} \approx 10\log(N_{\text{FH}}) \tag{A15}$$

where $N_{\text{FH}}$ denotes the number of frequency-hopping frequencies. The frequency-hopping spread spectrum gain for FH-BOC $(\alpha_{M-1}{:}1{:}\alpha_0, \beta)$ can be expressed as

$$\begin{aligned}G_{\text{FS}} &\approx 10\log\left[2\left(\frac{\alpha_{M-1}-\alpha_0}{\beta}+1\right)\right] \\ &= 10\log\left(\frac{\alpha_{M-1}-\alpha_0}{\beta}+1\right) + 3\end{aligned} \tag{A16}$$

Here, the term inside the $\log[\cdot]$ denotes the number of frequency-hopping frequencies for the two sidebands of the FH-BOC signal. Similarly, for the IFH-BOC $[\boldsymbol{\alpha}, \boldsymbol{\beta}]$:

$$\begin{cases} \boldsymbol{\alpha} = \left[\begin{array}{c}\boldsymbol{\alpha}^L \\ \boldsymbol{\alpha}^U\end{array}\right] = \left[\begin{array}{ccc}\alpha_{N-1}^L & 1 & \alpha_0^L \\ \alpha_{M-1}^U & 1 & \alpha_0^U\end{array}\right] \\ \boldsymbol{\beta} = \beta \times \mathbf{I}_{2\times1}\end{cases} \tag{A17}$$

The frequency-hopping spread spectrum gain can be expressed as

$$G_{\text{FS}} \approx 10\log\left(\frac{\alpha_{N-1}^L - \alpha_0^L}{\beta} + \frac{\alpha_{M-1}^U - \alpha_0^U}{\beta} + 2\right) \tag{A18}$$

Here, the term inside the $\log(\cdot)$ denotes the number of frequency-hopping frequencies for the two sidebands of the IFH-BOC signal.

By combining (A13), (A16), and (A18), the processing gain for FH-BOC and IFH-BOC can be derived:

$$G_{\text{FH}-\text{BOC}} \approx 10\log(\beta\frac{f_0}{f_D}) + 10\log(\frac{\alpha_{M-1} - \alpha_0}{\beta} + 1) + 3 \tag{A19}$$

$$G_{\text{IFH}-\text{BOC}} \approx 10\log(\beta\frac{f_0}{f_D}) + 10\log(\frac{\alpha_{N-1}^L - \alpha_0^L}{\beta} + \frac{\alpha_{M-1}^U - \alpha_0^U}{\beta} + 2) \tag{A20}$$

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
