# Peer review of "Frequency-Hopping Binary Offset Carrier Modulation with Independent Frequency-Hopping Patterns in Lower and Upper Sidebands"

_remotesensing, doi:10.3390/rs16224151_

Round 1

Reviewer 1 Report

Comments and Suggestions for Authors

Dear authors, many thanks for your manuscript. Please find attached my comments and suggestions to improve the manuscript's quality.

Reviewer 2 Report

Comments and Suggestions for Authors

General comments

This study introduces a novel modulation scheme, referring as independent frequency-hopping binary offset carrier(IFH-BOC), which includes three types and advances the concept of FH-BOC. IFH-BOC provides similar time-frequency characteristics and tracking performance under equivalent modulation parameters, while also offers enhanced anti-interference capabilities due to the distinct frequency-hopping patterns. This characteristic opens up more L-band frequencies for satellite navigation, giving signal designers greater flexibility for adjustments and optimizations.  The results is meaningful for future GNSS signal design.

Some specific issues should be addressed before it can be published.

Specific comments

1)   In Section 3, the authors defines three types of IFH-BOC, but the definitions are not used in Section 4 and Section 5. Moreover, the difference or merit of three different types have not been discussed. It is confused that why the authors define the different types of IFH-BOC.

It is suggested to discuss the performances of three types of IFH-BOC in Section 4, using the definitions, such as Type-I IFH-BOC, Type-II IFH-BOC, which also can be identified in Table 1.

For S1-S6, it is suggested to give a specific conclusion that which type is the best. And it should be clearly described in Section 6 “Conclusion”.

2) In “Abstract”, full descriptions of IFH-BOC and CEM should be given.

3)  The colors of S1 and S6 are too close, it is difficult to distinguish. It is better to change a color for S1 or S6.

4)  GNSSs should be GNSS in the text, such as in Line 35, Line 42, Line 157.

5)  Figure 6, the legend exceeds the axis of figure.

6)  Writing need to be improved.

Comments on the Quality of English Language

Writing can be improved.

Round 2

Reviewer 1 Report

Comments and Suggestions for Authors

All comments have been addressed appropriately, and the revisions have been implemented correctly. Congratulations on your very well-done work. There are no further comments from my side.